# Prognostic Effect of Comorbid Disease and Immune Gene Expression on Mortality in Kidney Cancer—A Population Based Study

**DOI:** 10.3390/cancers12061654

**Published:** 2020-06-22

**Authors:** Chung-Shun Wong, Tzu-Ting Chen, Wei-Pin Chang, Henry Sung-Ching Wong, Mei-Yi Wu, Wirawan Adikusuma, Yuh-Feng Lin, Wei-Chiao Chang

**Affiliations:** 1Graduate Institute of Clinical Medicine, College of Medicine, Taipei Medical University, Taipei City 110, Taiwan; johnson7617@gmail.com; 2Department of Emergency Medicine, Taipei Medical University-Shuang Ho Hospital, New Taipei City 23561, Taiwan; 3Department of Emergency Medicine, School of Medicine, College of Medicine, Taipei Medical University, Taipei City 110, Taiwan; 4Center for Neuropsychiatric Research, National Health Research Institutes, Miaoli County 35053, Taiwan; uniquepapa@gmail.com; 5School of Health Care Administration, College of Management, Taipei Medical University, Taipei City 110, Taiwan; wpchang@tmu.edu.tw; 6Department of Clinical Pharmacy, School of Pharmacy, Taipei Medical University, Taipei City 110, Taiwan; miningyue@gmail.com (H.S.-C.W.); adikusuma28@gmail.com (W.A.); 7Institute of Epidemiology and Preventive Medicine, College of Public Health, National Taiwan University, Taipei City 100, Taiwan; e220121@gmail.com; 8Division of Nephrology, Department of Internal Medicine, School of Medicine, College of Medicine, Taipei Medical University, Taipei City 110, Taiwan; 9Division of Nephrology, Department of Internal Medicine, Shuang Ho Hospital, Taipei Medical University, New Taipei City 23561, Taiwan; 10TMU Research Center of Urology and Kidney, Taipei Medical University, Taipei City 110, Taiwan; 11Department of Pharmacy, Faculty of Health Science, University of Muhammadiyah, Mataram 83115, Indonesia; 12Department of Internal Medicine, School of Medicine, College of Medicine, National Defense Medical Center, Taipei City 11490, Taiwan; 13Master Program in Clinical Pharmacogenomics and Pharmacoproteomics, School of Pharmacy, Taipei Medical University, Taipei City 110, Taiwan; 14Integrative Research Center for Critical Care, Wan Fang Hospital, Taipei Medical University, Taipei City 116, Taiwan; 15Department of Medical Research, Shuang Ho Hospital, Taipei Medical University, New Taipei City 23561, Taiwan

**Keywords:** kidney cancer, renal cell carcinoma, comorbidity, immune metagenes, survival

## Abstract

The effect of comorbidities and the immune profiles of the kidney cancer microenvironment play a major role in patients’ prognosis and survival. Using the National Health Insurance Research Database (Taiwan), we identified patients aged >20 years with a first diagnosis of kidney cancer between 2005 and 2014. Differences in demographic characteristics and comorbidities were examined using the Pearson chi-squared test or the t test. The Cox regression model was used to construct the nomogram. RNA-seq data were applied from The Cancer Genome Atlas database, and correlations between immune metagenes and clinical characteristics were determined using a linear regression model. In this nationwide cohort study, including 5090 patients with kidney cancer, predictors in our prediction models included age, sex, chronic kidney disease, dialysis requirements, renal stones, cerebrovascular disease, and metastasis tumor. In the tumor tissue profiles, significant positive correlations between immune metagenes and clinical stage or overall survival were observed among Natural Killer (NK) cells (CD56−), CD4+ T-helper 2 (Th2) cells, and activated Dendritic Cell (aDC). A negative correlation was observed between expression level of Dendritic Cell (DC) and overall survival. Patients with kidney cancer exhibit high prevalence of comorbid disease, especially in older patients. Comorbid disease types exert unique effects, and a particular comorbidity can affect cancer mortality. Moreover, the expression of immune metagenes can be utilized as potentialbiomarkers especially for further study of molecular mechanisms as well as microenvironments in kidney cancer.

## 1. Introduction

Renal cell carcinoma (RCC) is a common urinary malignancy that accounts for approximately 3% of all malignant tumors. The incidence rates vary substantially worldwide and are generally high in Europe and North America and low in Asia and South America [1]. The incidence has been steadily rising over the past three decades This may be attributed to the more liberal use of diagnostic imaging and early diagnosis [2]. Despite early diagnosis, metastasis is found in approximately 15% of patients with RCC at initial presentation [3]. The natural history of RCC is highly unpredictable. Some small renal tumors, which have distant metastases through the early haematogenic dissemination, are associated with a high risk of disease-specific mortality, whereas some patients with locally advanced renal tumors experience long-term disease-free survival [4].

Comorbidity is defined as any coexisting disease or condition that may affect the diagnosis, treatment, and prognosis of an index disease. The prognostic effect of comorbidity has been demonstrated in various cancers, including colon, lung, and prostate cancers [5]. Several models for the prediction of prognosis in patients with RCC have been reported [6,7,8,9]. Laboratory abnormalities, tumor-node-metastasis (TNM) stage, nuclear grade, and histologic subtype have been assessed as potential prognostic factors, but limited data are available on comorbidity as a prognostic factor for kidney cancer. The effect of comorbidities is crucial for patients’ survival, and gaining an enhanced understanding of how chronic conditions affect the overall survival of patients with RCC could aid in determining a patient’s prognosis and help in guiding treatment decisions.

The aim of this study was to quantify comorbidities and associations with overall survival in patients with kidney cancer. We created a prediction model for overall survival in patients with kidney cancer by using a Cox model. The assessment of statistical prediction model performance is essential; therefore, we demonstrated the performance of our prediction model through discrimination and calibration. Furthermore, we explored correlations between the expression of immune metagenes and clinical characteristics in patients with kidney cancer to understand the immune profiles of kidney cancer microenvironments.

## 2. Results

### Main Findings

Figure 1 shows the study design and participants. Patients exhibiting kidney cancer between 1 January 2005 and 31 December 2015 were identified (*N* = 8964). We excluded patients who had been diagnosed with kidney cancer before 2006 or after 2014, whose age or sex information was missing, whose age was <20 years, who did not receive nephrectomy, or who died before their application for catastrophic illness was approved. This yielded 5090 patients with kidney cancer. The basic characteristics, comorbid conditions, and Charlson comorbidity index were calculated [10,11]. Participants in this study are shown in Table 1. The mean age of patients with kidney cancer for the mortality group and the survival group was 65.89 ± 12.68 and 58.93 ± 13.05 years, respectively. More men than women were identified. The most prevalent comorbidities were hypertension (58.9% vs. 53.7%), diabetes mellitus (DM) (30.1% vs. 23.3%), metastatic tumor (26.2% vs. 5.1%), chronic kidney disease (CKD) (24.5% vs. 14.7%), peptic ulcer disease (15.7% vs. 12.9%), dialysis requirements (12.5% vs. 5.8%), chronic pulmonary disease (11.3% vs. 6.4%), and cerebrovascular disease (9.6% vs. 5.0%).

Clinical variables, namely, age, sex, DM, CKD, dialysis requirements, polycystic kidney disease (PKD), renal stones, myocardial infarction (MI), congestive heart failure (CHF), cerebrovascular disease, dementia, chronic pulmonary disease, peptic ulcer disease, and metastatic tumor were analyzed and statistically significant variables, including age, male sex, chronic kidney disease, dialysis requirements, renal stone, cerebrovascular disease, and metastasis tumor, were further selected and analyzed by using the cox regression model.

From the Cox model with stepwise selection, age (hazard ratio [HR]: 1.04 [1.04,1.05]), male sex (HR: 1.20 [1.06,1.36]), CKD [HR: 1.32 [1.10,1.58]), dialysis requirements (HR: 2.14 [1.70,2.71]), renal stones (HR: 1.55 [1.24,1.93]), cerebrovascular disease (1.37 [1.12,1.67]), and metastasis tumor (HR: 5.75 [5.04,6.57]) were statistically significant predictors for mortality in patients with kidney cancer (Table 2). Moreover, a nomogram based on these predictors was constructed (Figure 2). In clinical applications, the nomogram provides an easier means for calculating and interpreting survival rates for kidney cancer. In the nomogram, the HR for each predictor was transformed into point-based. Total risk scores were the sum of a patient’s individual risk score for each predictor, and these were used to predict each individual’s 1-, 3-, and 5-year survival (Figure 2), e.g., using the point scale in the nomogram, an 80 year-old (75 points) kidney cancer male patient (5 points) with comorbidities of cerebrovascular disease (10 points), without renal stone (0 point), chronic kidney disease (0 point), dialysis requirement (0 point), or metastatic tumor (0 point). The total sum of points was 90. The predicted 1-, 3-, and 5-year survival rates were 85%, 70%, and 60%, respectively. We examined the model performance through discrimination. The C-statistic is a widely applicable measure of predictive discrimination [12]. The overall Harrell’s C-statistic and Uno’s C-statistic were 0.74 and 0.72, respectively. The time-dependent AUC of the prediction model is shown in Figure 3. The integrated time-dependent AUC is 0.77. In addition, Figure 4 depicts the time-dependent receiver operating characteristic (ROC) curves at 1, 3, and 5 years. The AUCs were 0.75, 0.72, and 0.73, for 1, 3, and 5 years, respectively.

We also assessed the model performance through calibration, which describes the agreement between observed outcomes and predictions. The calibration plots of the model for 1-, 3-, and 5-year survival prediction demonstrated a reasonable fit, and the slope of the curve was close to 45-degrees, indicating ideal performance (Figure 5). Finally, to understand the immune profiles of the kidney cancer microenvironment, we explored correlations between the expression of immune metagenes and clinical characteristics in patients with kidney cancer by using the TCGA database (Figure 6A). As shown in Figure 6B, immune metagenes, including eosinophils, mast cells, and dendritic cells (DCs), provided a negative correlation to several clinical characteristics such as disease stage (Figure 6B and Appendix A); CD4+ Th1 cells, NK (CD56−) cells, CD4+ Th2 cells, NK cells, aDC, and pDC showed positive correlations to several clinical characteristics (Figure 6B and Appendix A). We revealed significant positive correlations of NK (CD56−) cells, CD4+ Th2 cells, and aDC with overall survival (OS), whereas a negative correlation was identified between DC and overall survival. These results suggest that immune profiles are involved in the development of kidney cancer. Thus, immune metagenes expression might be utilized as potential biomarkers for the progression of kidney cancer.

## 3. Discussion

In this nationwide cohort study, including 5090 patients with kidney cancer, we created a prediction model and point-based nomogram for OS in patients with kidney cancer. The predictors in this prediction model included age, sex, CKD, dialysis requirements, renal stones, cerebrovascular disease, and metastasis tumor. Our model predicted that 1-, 3-, and 5-year OS with AUCs were 0.75, 0.72, and 0.73, respectively. Moreover, calibration plots also showed a reasonable fit for the prediction of 1-, 3-, and 5-year survival.

Accurate prediction for kidney cancer survival is crucial for patient counseling, follow-up, and treatment planning. The TNM-derived American Joint Committee on Cancer classification represents the gold standard staging scheme for kidney cancer and is currently used for survival prediction [13,14]. However, several reports have highlighted the discrepancy between kidney-cancer-specific risk of death and the competing risk from comorbidities. Not all patients with kidney cancer die as a result of kidney cancer: Van Poppel et al. reported that 10.3% of patients with kidney cancer die as a result of kidney cancer, compared with 89.7% from other causes [15]. Lane et al. reported that the most common causes of death were cardiovascular (29%), rather than cancer progression (4%) [16]. Most comorbidities may be assumed to increase mortality rates compared with individual diseases because of the combination of multiple pathological processes simultaneously present. Therefore, comorbidities play a crucial role in survival of patients with kidney cancer and may also alter the risk of kidney-cancer-specific mortality.

Previous studies have assessed the association of the Charlson Comorbidity Index (CCI) with the OS of patients with kidney cancer [5,17], but this can be burdensome to compute and apply clinically. Additionally because CCI is an index of multiple factors, it cannot be used directly to identify which of the index components are associated with lower overall survival of an index disease. [18,19]. Our study quantified the effects of individual comorbidities among patients with kidney cancer and demonstrated the significant association of CKD, dialysis requirements, renal stones, cerebrovascular diseases, and metastatic tumor with OS.

In current study, we found that renal stones, CKD, and dialysis exerted significant HR for mortality in patients with kidney cancer. Renal stones are believed to increase the risk of cardiovascular events, CKD, and end-stage renal disease; a higher mortality rate may also be expected [20,21]. The exact mechanism driving the increased risk of cancer mortality resulting from renal stones is still unclear. One speculated pathogenic mechanism is a contribution by local irritation and inflammation to chronic systemic inflammation and cytokine release, which promote tumorigenesis [22]. CKD and cancer have been established to affect each other positively and negatively: cancer can cause CKD either directly or indirectly through the adverse effects of therapies, and CKD, conversely, may be a risk factor for cancer; both may be associated because both diseasesshare common risk factors [23]. Moreover, we demonstrated that patients with kidney cancer with CKD or dialysis requirements exhibited a higher risk of mortality, as reported by other studies [24,25]. The association of CKD with higher rates of cardiovascular diseases or other comorbid conditions may shorten survival and also directly impair survival, as for patients without cancer.

Our study also found that cerebrovascular diseases exerted significant HR for mortality in patients with kidney cancer. The association between an increased risk of cancer and increased thrombosis is commonly known. Tumor cells promote a hypercoagulable state and activate a clotting cascade through tumor procoagulants such as tissue factors, cancer procoagulants, and tumor mucins [26]. Hypercoagulability can also precipitate tumor growth or metastasis progression [27]. Therefore, venous thromboembolism negatively affects survival in patients with cancer [28,29].

Because the prevalence of both cancer and noncancer comorbidities increases with age, the integration of comorbidities into a competing-risk predictive model is essential [30,31]. Therefore, the strength of this study is ito quantify the effects of individual comorbidities among patients with kidney cancer. The large sample size (*N* = 8964) and high validity of cancer diagnoses gave more reliable conclusion. However, several potential limitations to this studyshould be addressed: First, we could not confirm the histological types and clinical stages of kidney cancer because the data from NHIRD contains no information on histological types and cancer staging, and these potential confounders may be associated with the risk of mortality. Second, comorbidity diagnosis based on ICD-9-CM codes may be less accurate than that from a complete interview, laboratory data, and clinical information. To validate comorbidity diagnoses in this study, we selected only patients who consecutively received diagnoses with specific comorbidity from clinical physicians at least three times. However, the study did not consider patients whose diagnoses were miscoded.

To address possible molecular mechanisms involved in the disease progression as well as in patient survival, RNA sequencing data were used to analyze immune metagenes for the clinical characteristics of patients with kidney cancer. We noticed a significant association between the immune profiles of tumor tissues and patients’ clinical characteristics. In particular, the immune infiltration of CD4+ Th2 cells and dendritic cells are associated with OS. Several studies have proposed immune infiltration to play critical roles in cancer progression and therapeutic effects. Ghatalia et al. showed higher aDC levels to be correlated with the recurrence of clear-cell RCC [32]. Consistent with previous findings, we observed that high levels of DC and aDC expression were associated with poor survival. In addition, Bindea et al., also suggested DC to be one of the risk prognostic factors in patients with chromophobe renal cell carcinoma [33], which fits nicely to our findings. Furthermore, we noticed that most studies addressing the roles of immune profiles are plagued with numerous uncertainties. We attribute this to the limited clinical sample size and the complex network of tumor microenvironments. Additional functional studies are required to evaluate the influence of DC in the carcinogenesis of kidney cancer.

## 4. Materials and Methods

### 4.1. Longitudinal Health Insurance Database

In this study, we obtained patient data from the National health Insurance Research Database (NHIRD), derived from Taiwan’s National Health Insurance (NHI) program. The NHI program in Taiwan was launched on 1 March 1995, and covered 99.9% of Taiwan’s population by 2014. The database contains detailed patient information, including data on sex; date of birth; residential or work area; dates of clinical visits; the International Classification of Diseases, Ninth Revision, Clinical Modification (ICD-9-CM) codes; prescription details; expenditure amounts; and outcomes at hospital discharge (recovery, death, or transfer). The NHIRD could also be linked to the database for cause of death to obtain information regarding the main cause of the patient’s death.

### 4.2. Study Design and Patient Population

The study cohort comprised all patients diagnosed with malignant neoplasm of the kidney (ICD-9-CM code: 189.0) between 1 January 2005 and 31 December 2014. Exclusion criterion were age <20 years; kidney cancer diagnosis before 1 January 2006; and no surgical treatment or death before approval of their application for catastrophic illness was obtained. Patients with missing variables, such as birth date and sex, were also excluded from the study. We identified the outcome of all-cause mortality, and each patient was then individually followed up from the index ambulatory visit until death or 31 December 2015. The study was approved after full review by the Joint Institutional Review Board of Taipei Medical University (TMU-JIRB N201912036). The study was conducted in accordance with approved guidelines. The informed consent of study participants was not required, because the dataset used in this study consisted of de-identified secondary data from Taiwan’s NHI program.

### 4.3. Measurement of Covariates and Comorbidities

Clinical variables, including age, sex, and date of kidney cancer diagnosis, were retrieved from the NHIRD. We also selected common chronic diseases that are likely to be associated with kidney cancer, for evaluation. Using ICD-9-CM codes, we identified all patients’ comorbidities at least three times during visits to the outpatient clinic or emergency department or once during their admission to the hospital one year prior to the index date. Comorbidities, including diabetes mellitus (DM) (ICD-9-CM code 250.X), chronic kidney disease (CKD) (ICD-9-CM codes 580-589), dialysis requirements, polycystic kidney disease (PKD) (ICD-9-CM codes 753.12 and 753.13), myocardial ischemia (MI) (ICD-9-CM codes 410 and 412), congestive heart failure (CHF) (ICD-9-CM codes 428, 398.91 and 402.x1), cerebrovascular disease (ICD-9-CM codes 430-438 and 362.34), dementia (ICD-9-CM codes 290.X, 294.1 and 331.0-331.2), chronic pulmonary disease (ICD-9-CM codes 490-496, 500-505 and 416.8.-416.9), peptic ulcer disease (ICD-9-CM codes 531-534), and metastatic tumor (ICD-9-CM codes 196-199), were analyzed. We defined renal stones in this study by using procedure codes for surgery one year prior to the index date.

### 4.4. Analysis of Immune Metagenes

RNA-seq data of 245 renal papillary cell carcinoma (KIRP) were downloaded from The Cancer Genome Atlas (TCGA) database with expression values of genes quantified in transcripts per kilobase million. The expression values of gene i were then transformed using Ei = log2(TPMi + 1) for downstream analyses. Gene signatures for 23 immune metagenes were adopted from a publication (Appendix A) [34], and the probe IDs were converted to gene symbols by using the R hgu133a.db package. Each metagene was calculated from signature genes by using the single-sample gene set enrichment analysis (ssGSEA) algorithm implemented in the R GSVA package. In details, given a metagene, we used the expression values of corresponding signature genes to construct the metagene score in KIRP samples. The expression values were input into ssGSEA algorithm, which output a gene set enrichment score (i.e., metagene score) per sample. Notably, ssGSEA algorithm gave the output value by calculating normalized difference of ECDFs (empirical cumulative distribution functions) of ranks of the genes that belong to and not belongs to given signature genes. The metagene score was then treated as an inferred quantity of corresponding infiltrated immune cells in each KIRP specimen. Correlation between (normalized) immune metagenes scores (as dependent variables) and clinical characteristics (as independent variables), which included depth of tumor invasion (T; *n* = 165 T1 vs. 25 T2 vs. 51 T3 vs. 2 T4), lymph node metastasis (N; *n* = 41 N0 vs. 20 N1 vs. 4 N2), distant metastasis (M; *n* = 90 no vs. 9 yes), and pathological stage (*n* = 156 Stage I vs. 18 Stage II vs. 45 Stage III vs. 14 Stage IV), was conducted using a linear regression model by adjusting age at diagnosis and gender. The direction of correlation (positive or negative) was defined using beta-coefficients (>0 or <0, respectively) from linear regression models. In addition, correlations between overall survival and immune metagenes were inspected by conducting a Cox-proportional hazard model by including age at diagnosis and gender of KIRP patients as covariates. The direction of correlation (positive or negative) was defined using hazard ratios (>1 or <1, respectively) from Cox regression models.

### 4.5. Statistical Analysis

We compared demographic data and comorbidities between the survival and mortality groups. Differences in demographic characteristics and comorbidities were examined using the Pearson chi-squared test or the t test. Clinical variables, namely, age, sex, DM, CKD, dialysis requirements, PKD, renal stones, MI, CHF, cerebrovascular disease, dementia, chronic pulmonary disease, peptic ulcer disease, and metastatic tumor, were analyzed. The time between entry to a study and a subsequent event and age were taken as continuous variables, and sex and the comorbidities, such as DM, CKD, dialysis requirements, PKD, renal stones, MI, CHF, cerebrovascular disease, dementia, chronic pulmonary disease, peptic ulcer disease, and metastatic tumor, were defined as categorical variables. We used stepwise selection approach, and then we obtained the final model with statistically significant variables as predictors for survival of kidney cancer. Finally, we used a Cox regression model to construct the nomogram.

We next compared the performance of the proposed prediction model by using the following measures from the Cox regression model. First, the time-dependent receiver operating characteristic (ROC) curves and area under the curve (AUC) characterize how accurately the fitted model can distinguish between individuals who experience an event and individuals who do not. Time-dependent ROC curves and AUC summarize predictive accuracy at specific times. Second, the concordance statistic (C-statistic) can be calculated as the proportion of pairs of individuals for which observed and predicted outcomes agree (are concordant) among all possible pairs in which one individual experienced the outcome of interest and the other did not. C-statistics provide overall measures of predictive accuracy. Finally, calibration plots were generated to explore the performance characteristics of the nomogram at 1-, 3-, and 5-year survival in patients with kidney cancer.

All statistical tests were two-sided, and P values less than 0.05 were considered statistically significant. Analyses were performed using SAS (SAS System for Windows, Version 9.4, SAS Institute Inc.,Cary, NC, USA) and R (R version 3.6.0 for Windows with packages Hmisc and rms, R Foundation for Statistical Computing, Vienna, Austria).

## 5. Conclusions

In conclusion, patients with kidney cancer exhibit high prevalence of comorbid disease, particularly older patients. Comorbid disease types exert unique effects, and particular comorbidities can affect cancer mortality. We constructed a comorbidity-based model to predict the 1-, 3-, and 5-year kidney cancer mortality, and this simple points-based tool may focus on specific comorbidities and their effects in clinical treatment decisions and long-term surveillance of patients with kidney cancer and with comorbid disease. Immune metagenes’ expression can also be utilized as potential prognostic and predictive biomarkers for patients with kidney cancer.

## Figures and Tables

**Figure 1 cancers-12-01654-f001:**
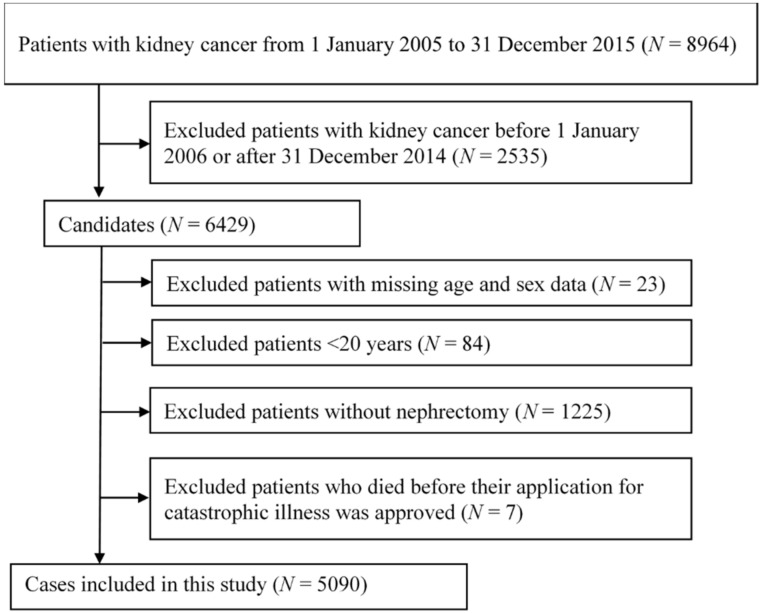
Selection criteria and process for eligible patients.

**Figure 2 cancers-12-01654-f002:**
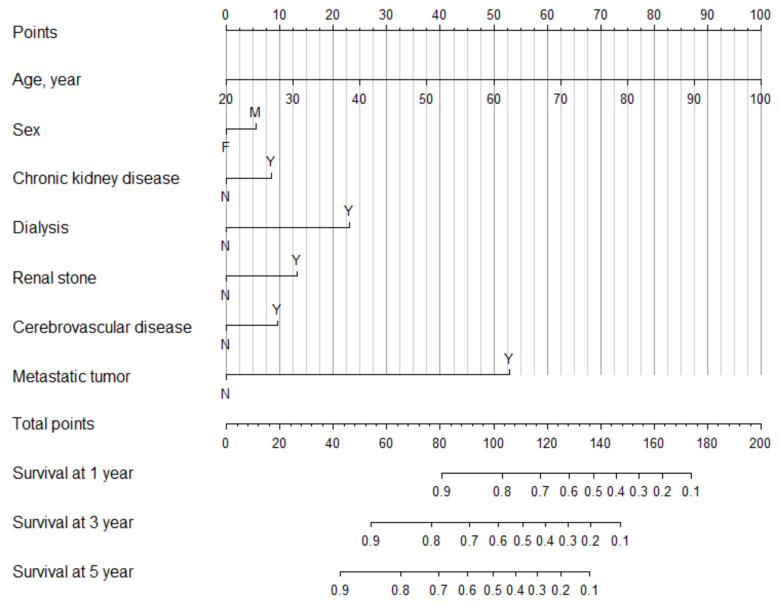
Nomogram prediction for 1-, 3-, and 5-year survival of kidney cancer. This nomogram tool is a direct representation of cox model given in Table 2. (Note: For a given patient profile, each predictor has a corresponding risk score (0–100) on the top “Points” scale. The risk scores from each predictor are summed to obtain a total point value. The total points are then indicated on the Total Points Scale (forth from the bottom). Finally, the corresponding predicted 1-, 3-, and 5-year survival is determined by drawing a vertical line down from the total points scale to the Predicted Survival Probability scale at 1, 3 and 5 years (the bottom three scales).

**Figure 3 cancers-12-01654-f003:**
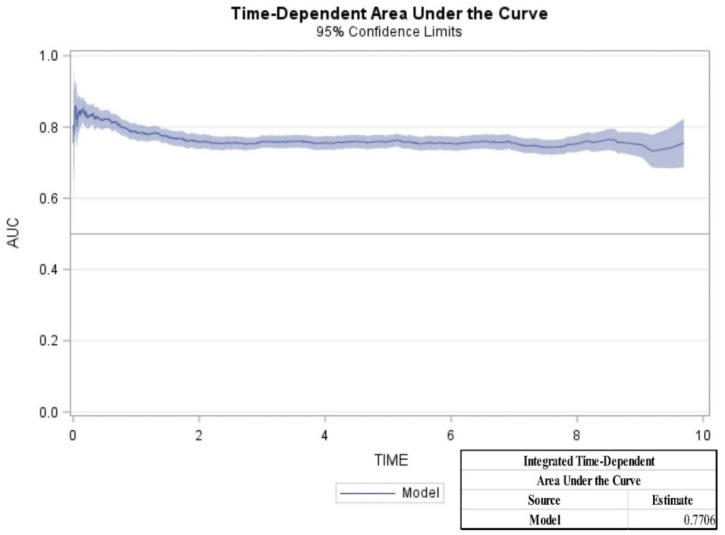
The integrated time-dependent area under receiver operating characteristic (ROC) curve and the 95% confidence limits for the predication model.

**Figure 4 cancers-12-01654-f004:**
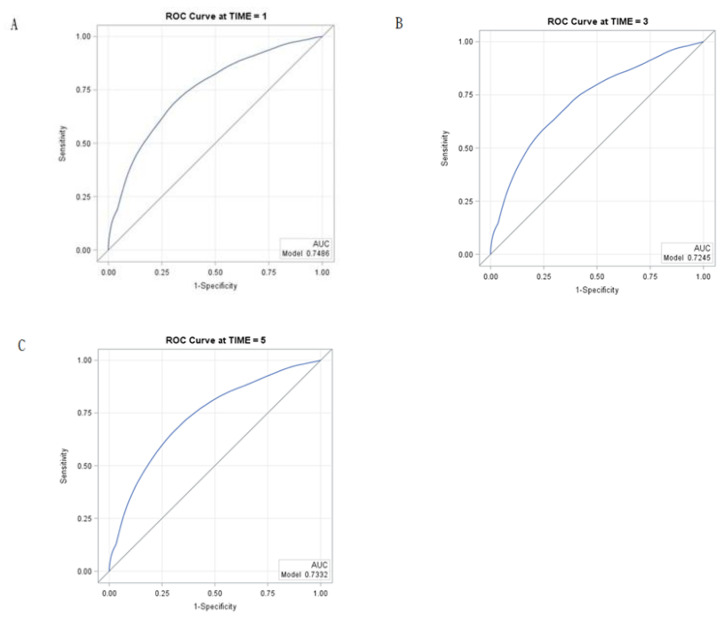
Time-dependent ROC curves at (**A**) 1 year, (**B**) 3 years, and (**C**) 5 years, to assess predictive accuracy for overall survival. ROC, receiver operating characteristic; AUC, area under curve.

**Figure 5 cancers-12-01654-f005:**
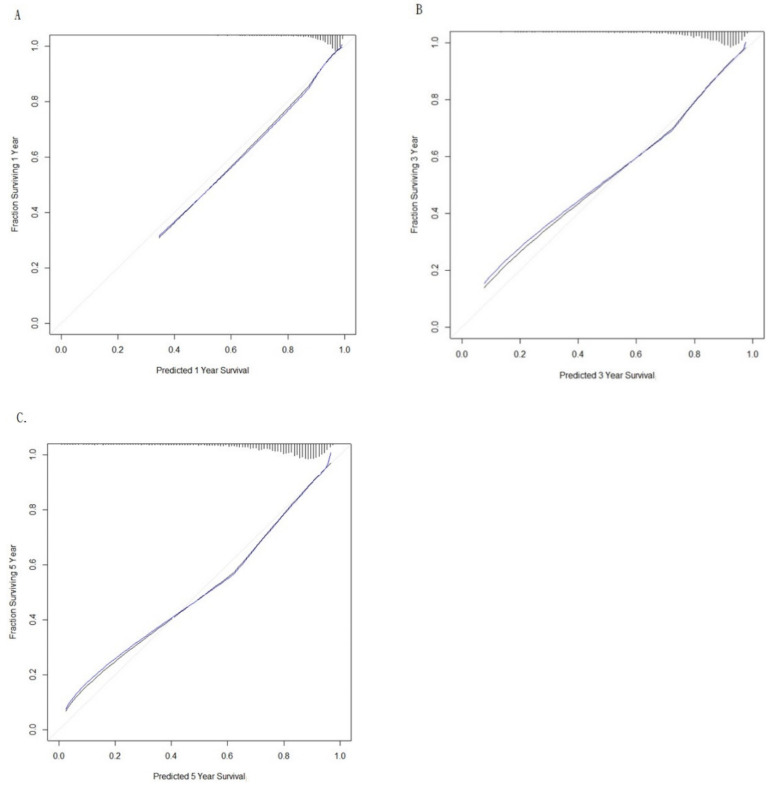
Kidney cancer overall survival nomogram calibration plots: (**A**) 1-year, (**B**) 3-year and (**C**) 5-year nomogram calibration curve. Y axis represents observed probability. X axis represents predicted probability. The predicted and observed probabilities of survival are graphed on the horizontal and vertical axes, respectively. The grey line indicates the reference line, on which an ideal model would lie. The black line represents observed, and blue line represents optimism-corrected, predictions.

**Figure 6 cancers-12-01654-f006:**
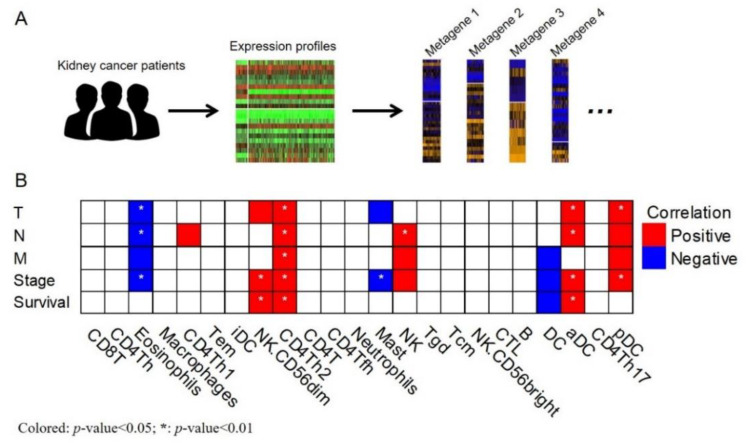
Correlation between immune metagenes and clinical characteristics. (**A**) Here is a schematic to explain the steps for construction of immune metagenes. In brief, expression values of renal papillary cell carcinoma (KIRP) patients were retrieved from The Cancer Genome Atlas (TCGA). For each category of 23 immune metagenes, single sample GSEA (ssGSEA) algorithm was adopted to calculate enrichment score (metagene score) of each patient/specimen. (**B**) Heatmap is the correlation results between immune metagenes and several clinical features (depth of invasion, T; lymph node metastasis, N; distant metastasis, M; pathological staging, stage; and overall survival). Color: red and blue for positive and negative correlation, respectively (without *: *p*-value < 0.05; with *: *p*-value < 0.01); white for nonsignificant (*p*-value > 0.05).

**Table 1 cancers-12-01654-t001:** Demographic and clinical characteristics of patients with kidney cancer from National health Insurance Research Database (Taiwan).

	Death	Survivor
*N*	1181	3909
Age, mean ± SD, y	65.89 ± 12.68	58.93 ± 13.05
Age group, *N* (%), y		
20–39	38 (3.2)	327 (8.4)
40–59	337 (28.5)	1754 (44.9)
60–79	657 (55.6)	1628 (41.6)
80–99	149 (12.6)	200 (5.1)
Male, *N* (%)	795 (67.3)	2515 (64.3)
Comorbid conditions for 1 year prior to the kidney cancer, *N* (%)		
Diabetes mellitus	356 (30.1)	910 (23.3)
Hypertension	696 (58.9)	2099 (53.7)
Chronic kidney disease	289 (24.5)	576 (14.7)
Dialysis requirements	148 (12.5)	228 (5.8)
Polycystic kidney disease	9 (0.8)	33 (0.8)
Renal Stone	87 (7.4)	220 (5.6)
Myocardial infarction	22 (1.9)	31 (0.8)
Congestive heart failure	86 (7.3)	152 (3.9)
Peripheral vascular disease	41 (3.5)	89 (2.3)
Cerebrovascular disease	113 (9.6)	195 (5.0)
Dementia	17 (1.4)	22 (0.6)
Chronic pulmonary disease	133 (11.3)	252 (6.4)
Peptic ulcer disease	185 (15.7)	504 (12.9)
Moderate or severe liver disease	5 (0.4)	11 (0.3)
Metastatic tumor	310 (26.2)	201 (5.1)
Charlson comorbidity index		
0–1	17 (1.4)	60 (1.5)
2	269 (22.8)	1688 (43.2)
3	208 (17.6)	976 (25.0)
4	161 (13.6)	510 (13.0)
≥5	526 (44.5)	675 (17.3)
mean ± SD	5.03 ± 2.94	3.30 ± 1.87
The year for newly diagnosis of kidney cancer. *N* (%)		
2006	124 (10.5)	217 (5.6)
2007	159 (13.5)	262 (6.7)
2008	178 (15.1)	354 (9.1)
2009	158 (13.4)	342 (8.7)
2010	135 (11.4)	372 (9.5)
2011	125 (10.6)	495 (12.7)
2012	119 (10.1)	549 (14.0)
2013	100 (8.5)	592 (15.1)
2014	83 (7.0)	726 (18.6)

Abbrevations: SD, standard deviation; *N*, number.

**Table 2 cancers-12-01654-t002:** Cox model-based analysis of significant prognostic factors, including age, gender, and comorbidities of kidney cancer patients.

Risk factor	Hazard Ratio	*P*-Value
Age, y	1.04 (1.04,1.05)	<0.0001
Male	1.20 (1.06,1.36)	0.0037
Chronic kidney disease	1.32 (1.10,1.58)	0.0023
Dialysis requirements	2.14 (1.70,2.71)	<0.0001
Renal stones	1.55 (1.24,1.93)	0.0001
Cerebrovascular disease	1.37 (1.12,1.67)	0.0018
Metastatic tumor	5.75 (5.04,6.57)	<0.0001

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
