# Peer review of "Prognostic Effect of Comorbid Disease and Immune Gene Expression on Mortality in Kidney Cancer—A Population Based Study"

_cancers, 2020, doi:10.3390/cancers12061654_

Round 1

Reviewer 1 Report

This is a well written paper that presents important information.  I have marked cases where minor corrections in the manuscript are needed. There is a floating legend in Figure 3 that needs to be placed. I recommend adding a example describing the use of the nomogram to the text. It has been a pleasure reviewing this manuscript. I am recommending it for publication.

Reviewer 2 Report

This manuscript describes a cohort study using database of kidney cancer patients. Comorbid diseases were found to have unique effects on mortality of kidney cancer patients. The expression of immune metagenes was also found to be potential prognostic and predictive biomarkers. This manuscript does not have enough data. Cancer stage is important but it’s missing.

Fig. 2 requires elaboration on how to read.

Clearly and concretely describe the results in the Abstract, Results and Conclusion sections

Were other metagene profiles explored?

In Abstract, “DC” requires a full name.

Reviewer 3 Report

The authors assessed the effect of comorbidities and the immune profiles of the kidney cancer microenvironment on patient's prognosis and survival.

Between different comorbidities the authors also mention diabetes mellitus, myocardial ischemia, congenistive heart failure as comorbidities. From the section Results I don't know directly what are resutlts of analysis of this data. Were these factors non-significant?

I don't see the diabetes mellitus data in the table 1.

The authors reported some limitations of the study - lack data on cancer staging. But the authors assessed metastatic disease as a risk factor.

Could the authors explain this issue?

Reviewer 4 Report

Based on my opinion, the following questions need to be addressed:    1. Sizable portion of 8964 patients (N=2535) retrieved from the registry were not included to the study, because they have been diagnosed during the years of 2005 or 2015. Why were these patients not included? Perhaps, the patients diagnosed during 2015 had too short follow-up time for the inclusion to the study, but this does not apply to the patients diagnosed in 2005.     2. More details about Cox model building needs to be provided: Which variables were included to the first model, what were the results and how they were itteratively removed? The assumption of proportional hazards is absolutely critical for applicability of this model and the proportionality of hazards needs to be validated by appropriate methods.    3. Table 1 compares counts and percentages for values of different variables between the groups of dead and surviving patients. Based on this table, a conclusion is provided that "mortality group had more comorbid conditions". To this reviewer, this approach does not seem appropriate, because these two groups have likely differed in their follow-up times. We do not know, based on the presented data, whether or not the patients in the mortality group had higher chance to develop the outcome (death), because they were followed for longer time, while the patients in survival group were followed for less person-years, and so they could not have developed outcome while they were in the study   4. Analysis does not consider or at least does not indicate in the Table I the distribution of histological types and clinical stages of kidney cancer. These variables are reasonably expected to influence the risk of death and they need to be controlled if the inference is being made about the effect of comorbidities. Because the use of Cox model is reported in a very limited way, this reviewer cannot see some important details on how this modeling was implemented, for instance how was the year of diagnosis modeled (continuous variable? categorical variable?).    5. It is indicated in the text that database allows to retrieve the cause of death information; however, in this study, death is not specifically defined, and it appears that the death of any cause is considered as an outcome. Analyses focusing on death of cancer and death attributable to non-cancer causes would be warranted, or at least these two different outcomes should be discussed more extensively, especially when the role of comorbidities on the mortality of kidney cancer patients is evaluated, because if the quality of data allows it, it would be good to know whether the higher risk of death of kidney cancer patients with comorbidities can be attributed to the progression of cancer or to the progression of comorbid diseases. Studies that addressed this question have been recently reported in high-profile journals and this question seems to be valid and worth addressing by this type of studies.   6. Overall, methodological description is extremely limited and lets a lot of guessing on the side of readers. For instance, determination of Charlson Comorbidity Index has been multiple times updated, and we have no information in the text which specific version/weights/comorbidity categories were used.  This insufficient reporting applies to the performance evaluation by time-dependent AUCs, but more prominently to the whole gene expression analysis. The authors did not provide information on metagenes whose "correlation" with clinical characteristics they investigated  using the linear regression model. This reviewer cannot assess whether this analysis was conducted in a proper way due to a very limited description of methods. What were the clinical characteristics in terms of variable type? Were these variables appropriate for linear regression? Also, without providing any background on immune metagenes, readers are forced to read full-text reference provided in this manuscript, and even this reference does not seem to provide enough information to understand this analysis. GSEA typically produces enrichment scores and p-values for enrichment of a biologically meaningful list of genes in a rank-ordered list of genes from a given experimental study (or set of studies). I am missing any information about the composition of metagenes; about specific studies that were retrieved for this analysis from the TCGA (how many studies, which IDs, how distributed clinical characteristics across cases, how was the GSEA performed/which settings were used in the GSEA analysis...). 

Round 2

Reviewer 2 Report

The authors have addressed some issues so this manuscript is acceptable.

Author Response

Thank you for your suggestion.

Reviewer 4 Report

The authors did not address all issues previously reported by this reviewer.  Among the issues not addressed by study authors, the most critical unresolved issue is the lack of justification for the use of proportional hazards model (Cox regression) for survival analysis. It was specifically requested by this reviewer that "The assumption of proportional hazards is absolutely critical for applicability of this model and the proportionality of hazards needs to be validated by appropriate methods.". Since the proportional hazards model is built entirely around this assumption, if it happens to be invalid for a set of predictors in a given dataset, then the Cox model should not be used on that dataset, and any results would be questionable. Test for proportionality of hazards was not provided, and so it is impossible to conclude whether the data met the assumption needed for this model. Authors stated that logistic regression seems less suitable, which is true, but this alone does not justify the use of Cox regression. Furthermore, the authors state that "The Cox proportional hazards model is the most popular model for the analysis of survival data"; however, popularity of any statistical method cannot replace validation of its underlying assumptions, which is prerequisite for the use of any statistical test or model for analysis of specific data. There are numerous variants of Cox proportional hazard model as well as other survival analysis methods besides Cox model, which can be used, if the assumption of proportionality of hazards is found invalid.    

Author Response

Thank you for your suggestions !